# Automatic Multiple Articulator Segmentation in Dynamic Speech MRI Using a Protocol Adaptive Stacked Transfer Learning U-NET Model

**DOI:** 10.3390/bioengineering10050623

**Published:** 2023-05-22

**Authors:** Subin Erattakulangara, Karthika Kelat, David Meyer, Sarv Priya, Sajan Goud Lingala

**Affiliations:** 1Roy J. Carver Department of Biomedical Engineering, University of Iowa, Iowa City, IA 52242, USA; subin-erattakulangara@uiowa.edu (S.E.); karthika-kelat@uiowa.edu (K.K.); 2Janette Ogg Voice Research Center, Shenandoah University, Winchester, VA 22601, USA; dmeyer2@su.edu; 3Department of Radiology, University of Iowa, Iowa City, IA 52242, USA

**Keywords:** dynamic speech MRI, articulator segmentation, protocol adaptiveness, transfer learning

## Abstract

Dynamic magnetic resonance imaging has emerged as a powerful modality for investigating upper-airway function during speech production. Analyzing the changes in the vocal tract airspace, including the position of soft-tissue articulators (e.g., the tongue and velum), enhances our understanding of speech production. The advent of various fast speech MRI protocols based on sparse sampling and constrained reconstruction has led to the creation of dynamic speech MRI datasets on the order of 80–100 image frames/second. In this paper, we propose a stacked transfer learning U-NET model to segment the deforming vocal tract in 2D mid-sagittal slices of dynamic speech MRI. Our approach leverages (a) low- and mid-level features and (b) high-level features. The low- and mid-level features are derived from models pre-trained on labeled open-source brain tumor MR and lung CT datasets, and an in-house airway labeled dataset. The high-level features are derived from labeled protocol-specific MR images. The applicability of our approach to segmenting dynamic datasets is demonstrated in data acquired from three fast speech MRI protocols: Protocol 1: 3 T-based radial acquisition scheme coupled with a non-linear temporal regularizer, where speakers were producing French speech tokens; Protocol 2: 1.5 T-based uniform density spiral acquisition scheme coupled with a temporal finite difference (FD) sparsity regularization, where speakers were producing fluent speech tokens in English, and Protocol 3: 3 T-based variable density spiral acquisition scheme coupled with manifold regularization, where speakers were producing various speech tokens from the International Phonetic Alphabetic (IPA). Segments from our approach were compared to those from an expert human user (a vocologist), and the conventional U-NET model without transfer learning. Segmentations from a second expert human user (a radiologist) were used as ground truth. Evaluations were performed using the quantitative DICE similarity metric, the Hausdorff distance metric, and segmentation count metric. This approach was successfully adapted to different speech MRI protocols with only a handful of protocol-specific images (e.g., of the order of 20 images), and provided accurate segmentations similar to those of an expert human.

## 1. Introduction

The human upper airway consists of oral, pharyngeal, and laryngeal structures that intricately coordinate to perform essential tasks such as breathing, swallowing, and speaking. These structures include the lips, tongue, hard palate, soft palate (or the velum), pharynx, epiglottis, and vocal folds. The coordination of these structures changes the vocal tract airspace (from the lips to the vocal folds), thus filtering the glottal sound source to create the speech phonemes (vowels and consonants). Analyzing the changes in vocal tract airspace and soft-tissue articulators (e.g., the tongue and velum) enhances our understanding of speech production. Magnetic resonance imaging (MRI) has emerged as a powerful modality for safely assessing the dynamics of the vocal mechanism during speech production [1,2,3]. It has several advantages over competing modalities, such as a lack of ionizing radiation compared to X-rays; higher soft-tissue contrast compared to ultrasound; the ability to visualize in flexible/arbitrary planes; and the ability to visualize deep structures in the vocal tract, such as the vocal folds, in contrast to electromagnetic articulography. Dynamic MRI has been used to better understand the phonetics of language [4,5], understand songs [6], beat-boxing [7], and understand speaker-to-speaker differences [5,8]. In addition, it has been used in clinical applications, such as assessing velopharyngeal insufficiency [9,10], assessing speech post-glossectomy [11,12], and assessing oropharyngeal function in subjects with cleft lip and palate [13]. A long-standing challenge with MRI has been associated with its slow acquisition speed, which reflects compromises among the achievable spatio-temporal resolutions, and vocal tract coverage. To address this, several “fast speech MRI” protocols have emerged in recent years [14,15,16,17,18]. These primarily operate by violating the Nyquist sampling criterion (i.e., they create images more quickly by acquiring fewer samples) and reconstructing by imposing constraints on the dynamic images. Several such protocols have emerged based on the choice of scanner field strength, sampling, and reconstruction constraints [3]. Notably, two recent open-source databases from these protocols have been released to the scientific community [8,19]. The size of the dynamic images has increased considerably with the high frame rates in the above-mentioned fast protocols. For example, a 15-second spoken utterance with an 83 frame/s protocol produces ~1245 image frames [14]. Unfortunately, manual annotation of vocal organs in these datasets is both labor-intensive and time-consuming. In our experience, expert voice users spend an average of 1–2 min segmenting an individual articulator in one frame. Thus, there is a need to automate this process, and a need for segmentation algorithms that can be generalized to different fast speech MRI protocols.

Upper-airway MRI segmentation methods may be classified either by the type of data being segmented (e.g., static 2D images, static 3D volumes, 2D dynamic images, or 3D dynamic volumes) or by the level of human intervention required, ranging from manual to semi-automatic to fully automatic methods. Several semi-automatic methods have been developed for 2D dynamic speech MRI segmentation. Bresch et al. used a semi-automatic approach, where a manually segmented vocal tract model was used as an anatomical template, and the parameters of this model were estimated for different dynamic frames via an iterative optimization algorithm where data consistency was enforced with the acquired k-space data [20]. This approach is computationally intensive, requiring ~20 min to segment a single frame, and every new dataset requires manual annotation. Kim et al. proposed a semi-automatic user-guided segmentation approach for 2D dynamic speech MRI, where the user specifies grid points along the vocal tract, and this approach provides air tissue boundary segmentation with image quality enhancement, pixel sensitivity correction, noise reduction, and airway path estimation [21]. For 3D segmentation, Javed et al. used seed-growing-based segmentation to identify the deforming 3D airway collapse in obstructive sleep apnea [22]. Similarly, Skordilis et al. used seed-growing segmentation to identify the tongue in 3D across various speech postures [23]. More recently, deep learning-based methods have been developed to automatically label and segment articulators in the upper airway [24,25,26]. Fully convolutional networks (FCNs) were used in [24] and [27] for segmenting airways. In [28], the authors segment multiple articulators, such as the upper lip, hard palate, soft palate, vocal tract, lower lip, jaw, tongue, and epiglottis, from dynamic 2D speech MRI. The training set in this study used real-time MRI data from five speakers with 392 images acquired on a 3T Philips Achieva MRI scanner and 16-channel neurovascular coil. [28] They used the conventional U-NET model and applied it to a segment of real-time speech MRI data from five speakers with 392 training set images acquired with a 3.0T Philips Achieva MRI scanner and 16-channel neurovascular coil. In [29], the authors developed a novel upper airway segmentation method using anatomy-guided neural networks for both static and dynamic MRI scans. In this generalized region of interest (GROI) approach, the airway was divided into different regions, and manual parcellation into static 3D MRI data was implemented. For dynamic 2D datasets, the method utilized 3440 annotated slices of mid-sagittal scans and employed the conventional U-NET model with modified loss as a function of false positive and false negative rates. One limitation of these algorithms is that it was not tested on different speech MRI acquisition protocols, and the classical U-NET model warrants the need for re-training with a large number of samples when switching to a new protocol.

In this paper, we propose a stacked transfer learning-based U-NET model to segment the deforming vocal tract airspace, the tongue, and the velum in dynamic mid-sagittal 2D speech MRI datasets. Our approach leverages (a) low- and mid-level features and (b) high-level features. The low- and mid-level features are derived from models pre-trained on labeled open-source brain tumor MR and lung CT datasets and an in-house airway labeled dataset. The high-level features are derived from labeled protocol-specific MR images. We developed three different networks to individually segment the vocal tract’s airspace, tongue, and velum. In this investigation, our approach was successfully adapted to different speech MRI protocols with only a handful of protocol-specific images (e.g., on the order of 20 images) and provided accurate segmentations similar to those of an expert human. The main contribution of this work is the development of the protocol-adaptive U-NET model based on transfer learning. This model effectively handled multiple fast-speech MRI protocols and required only a small amount of protocol-specific training data. Our contribution also includes the release of our segmentation and pre-processing codes as open-source software. Our model’s robustness and efficiency make it a valuable addition to the field of speech and voice science. 

## 2. Methods

Three networks are built to individually map the vocal tract airspace, tongue, and velum to their respective segmentations. We utilize a multi-level transfer learning scheme to hierarchically adapt the network to low/mid/high-level features from the various datasets. To learn the low- and mid-level features required for effective performance, our network was pre-trained with multiple annotated datasets. These were sourced from publicly available databases: one containing brain-MR scans and the other containing lung-CT scans. Additionally, we utilized an in-house annotated speech MRI database, as shown in Figure 1.

Following pre-training, we applied transfer learning to re-train the network and learn high-level features from three different protocol-specific dynamic speech MRI datasets. This transfer learning approach allowed the network to adapt to the unique characteristics of each dataset, enabling it to accurately segment vocal tract structures in dynamic speech MRI scans. The use of multiple datasets for pre-training and transfer learning helped to improve the overall robustness and accuracy of our network, making it better equipped to handle a wide range of dynamic speech MRI datasets. Finally, the model is applied to segment the vocal tract structures of interest from unseen dynamic images acquired from the specific protocol at hand. In the following sections, we describe the details of the datasets used, the transfer learning-based pre-training and re-training of STL U-NET, and the implementation of our network.

Brain Tumor segmentation dataset (BRATS): This database contains multi-modal brain tumor MR datasets, where regions of low- and high-grade glioma tumors were manually labeled in a variety of MR contrast images, including native T1-weighted, post-contrast T1-weighted, T2-weighted, and T2-FLAIR volumes [30,31]. We reformatted a total of 119 T2-weighted volumes in 1138 2D slices, with the goal of leveraging the low-level segmentation features from this T2-weighted labeled database via transfer learning in our network to segment the soft-tissue vocal tract structures (e.g., the tongue and velum). Our rationale for using the T2-weighted volumes was that the task of segmenting bright-intensity tumors from the surrounding darker-intensity healthy tissue shared similarities with the task of segmenting bright soft-tissue vocal tract structures from the surrounding dark airspace.

Thoracic CT segmentation dataset: This dataset was created in association with a challenge competition and conference session conducted at the American Association of Physicists in Medicine (AAPM) 2017 annual meeting [32]. It contained 60 patients with segmentations of multiple structures, such as the esophagus, heart, left and right lungs, and spinal cord. For our work, we used the right and left lung segments to learn the low-level features in the network that segments the vocal tract airspace. The 3D volumes and segmentations were sliced down into a total of 1582 2D slices, which were used as data instances for our purposes.

In-house dynamic 2D Cartesian speech MRI dataset: We collected dynamic speaking datasets on a 3T GE Premier Scanner with a custom 16-channel airway coil using a Cartesian 2D dynamic GRE sequence with the following parameters: FOV: 20 cm × 20 cm; 2.5 mm × 2.5 mm; slice thickness = 6 mm; flip angle = 5 degrees; parallel imaging acceleration factor = 2; temporal resolution = 153 ms; scan duration = 23 s. A total of 4 speakers were scanned. They were asked to produce a range of speaking tasks, such as producing alternating consonant and vowel sounds, “za-na-za, zee-nee-zee, zu-nu-zu, lu-lee-laa”, and fluent speech, such as counting numbers out loud. The subjects were asked to voluntarily slow their speaking rate to minimize the motion blurring of fast articulatory movement. Parallel imaging reconstruction was performed online, and images were exported as DICOM files. The vocal tract airspace was then manually segmented in the MATLAB environment by a trained biomedical engineer with 4 years of image processing experience (author: Erattakulangara). A total of 914 2D dynamic frames from data combined from all the speakers were manually segmented and were used for transfer learning the mid-level features in our network for airspace segmentation.

Fast speech MRI datasets: To demonstrate the protocol adaptiveness of the stacked UNET model, we used it to segment vocal organs in dynamic speech MR images obtained from three different fast-speech MRI protocols. The details of these protocols and the imaging data used in this study are detailed below. 

Protocol 1: In this protocol, we considered 2D mid-sagittal dynamic images from the open-source French speaker speech MRI database [19]. Data acquisition was performed on a Siemens 3T Prisma scanner with a 64-channel head-neck coil. Radial sampling was performed with an RF-spoiled FLASH sequence with sequence parameters, TR/TE = 2.2 ms/1.47 ms; FOV = 22 × 22 cm; flip angle = 5 degrees; slice thickness = 8 mm. Spatio-temporal resolutions were 1.6 mm and 20 ms/frame. Reconstruction was conducted via non-linear sparsity-based temporal constrained reconstruction with a joint estimation of coil sensitivity maps and dynamic images. In this study, we randomly chose dynamic image data from three speakers (1 for training, 1 for validation, and 1 for testing). A total of 20/3/10 dynamic image frames, respectively, from Speaker 1, Speaker 2, Speaker 3, were used in the training, validation, and testing sets. This database contained speech sounds from French speakers. Sixteen different French speech tokens were used in this protocol. To capture the maximum variation of the vocal tract motion, we randomly choose image frames from dynamic datasets across these speech tokens. 

Protocol 2: This protocol used images from the University of Southern California’s open-source multi-speaker raw MRI database [8]. Images were acquired on a GE 1.5 T Signa MRI scanner using an eight-channel custom vocal tract coil and a uniform-density spiral GRE sequence. TR was ~6 ms, readout duration = 2.4 ms; slice thickness was 6 mm; spatio-temporal resolutions were 2.4 × 2.4 mm^2^ and 12 ms/frame. Reconstruction was achieved via a sparse SENSE temporal finite difference iterative algorithm. Similar to protocol 1, we randomly choose three speakers from this database (one for training, one for validation, and one for testing). These speakers produced fluent speech, including speaking the “grandfather passage”, speaking sentences in the English language, and counting numbers out loud. As with protocol 1, we randomly chose dynamic images from these speaking tasks such that 20/3/10 images were respectively drawn from Speaker 1/2/3 and belonged to the training, validation, and testing sets. 

Protocol 3: This protocol was developed on the 3T GE Premier scanner at the University of Iowa. Data were collected from three speakers with a custom 16-channel upper-airway coil in accordance with the University of Iowa’s institutional review board policies. A variable density-based spiral GRE sequence was implemented with a TR of 6 ms; readout duration = 1.3 ms; slice thickness = 6 mm; spatio-temporal resolutions of 2.4 × 2.4 mm^2^ and 18 ms/frame. Reconstruction was achieved by a learning-based manifold regularization scheme. Speakers produced alternating consonant and vowel sounds and fluent speech by counting numbers out loud. As in the above protocols, 20/10/3 dynamic images, respectively, from Speaker 1, Speaker 2, Speaker 3, were used in training, validation, and testing sets.

Generation of manual segmentations: To train, validate, and test the STL UNET, we leveraged human annotators to generate manual segmentations. To reduce the subjectivity of test labels, each image in the test set was segmented by three expert humans. The first human annotator was a radiologist with expertise in body, cardiovascular, and thoracic imaging, and with more than 10 years of experience in radiology (author: Priya). The second human annotator was a professional voice user, researcher, and vocologist with more than 20 years of experience in human vocal tract anatomy and voice research (author: Meyer). The third annotator was a graduate student with 4 years of experience in upper-airway MRI and image processing (author: Erattakulangara). Prior to segmentation, all three users established and agreed to an anatomical guide and landmarks to segment the vocal tract (see Figure 2). The three users listed above manually annotated the test set in each of the three protocols by manual pixel-wise labeling of tongue, velum, and airway in either MATLAB R2021 (Mathworks, Natick, MA, USA) or the Slicer platforms (https://www.slicer.org) accessed on 4 August 2022. The third user alone annotated the images in the training and validation sets.

Pre-training and re-training of the STL U-NET: Figure 3 shows the STL U-NET Model with the three individual networks that respectively segment the tongue, velum, and vocal tract airspace. Note that while the network in Figure 3b is the standard U-NET model with the contracting and expanding blocks, it differs from how we train the model by leveraging the previously mentioned datasets. This model has a total of 39 layers, including the convolution, RELU, max-pooling, and concatenation steps. The two networks that segment the soft tissues (i.e., tongue and velum) were first trained with 1138 labeled T2-weighted brain tumor MR images from the open-source BRATS database. In a second step, the layers of the pre-trained models capturing the low-level features were frozen, and the model was re-trained with the 20 protocol-specific speech MR images to optimize the remaining m number of layers representing the high-level features. Similarly, the network segmenting the vocal tract airspace was first pre-trained with 1582 labeled lung CT images from the AAPM database, and 914 in-house dynamic speech MR images to learn both low- and mid-level features. The first layers with the low- and mid-level features were frozen, and the network was re-trained again to learn the high-level features in the remaining m number of layers using the 20 protocol-specific MR images. Figure 3b shows a schematic of how each of the U-NETs was re-trained with a protocol-specific dataset. We employed an exhaustive grid search optimization criterion to find the parameter m above in all the three networks based on achieving the least validation loss (see Table 1). We specifically swept through the parameter space of m spanning between 5 and 35, in steps of 5. Later, the step of 5 was fine-tuned to identify the layer number that yielded the lowest validation loss for each component: velum, tongue, and airway. Our results showed that the optimal layer numbers for these components were 14, 19, and 21, respectively.

The network was implemented using TensorFlow, and training was performed on an Intel Core-i7 8700CK, 3.70 GHz 12 core CPU machine. We used the following binary cross entropy loss function to train the U-NET model. Here y is the reference label and $p\left(y\right)$ is the predicted probability.
$$H_{p}\left(q\right)=-\frac{1}{N}\sum_{i=1}^{N}y_{i}\cdot log\left(p(y_{i}\right))+\left(1-y_{i}\right)\cdot log(1-p\left(y_{i}\right))$$

The network was trained with the Adam optimizer, which has hyperparameters *β*_1_ = 0.9, *β*_2_ = 0.999 and *ε* = 1 × 10^−8^. The training was conducted for 150 epochs using early stoppage criterion. Data augmentation was also performed on the input data to increase the number of protocol-specific speech MR training samples by a factor of 4. Rotation, scaling, and cropping were the basic operations performed to augment the images. For training, a crucial part of achieving the desired performance is tuning the hyperparameters. The parameters given in Table 1 are considered to be relevant to network performance. The second column shows the range of parameters we have tested to find the best among them. This grid-based search allows the network to test all the combinations of these parameters and to select the best based on the lowest validation loss.

Pre-processing and post-processing: In this study, we applied pre- and post-processing to improve the STL U-NET performance. First, data curation from the different speech MRI protocols was applied while re-training the STL U-NET model. Each protocol may contain images where the airway, tongue boundary, or velum boundary are poorly visualized due to swallowing, or images with motion artifacts and un-resolved alias artifacts. Figure 4 shows examples that were omitted from the training set. The input dynamic images from the protocols were cropped in pre-processing to focus on the regions of interest in the vocal tract and surrounding soft tissue vocal organs. Images were then re-scaled to a 256 × 256 image matrix size. Since the reconstructed images may contain non-uniform intensities, we applied a bias field intensity correction algorithm to provide uniform intensity across the field of view. In our earlier experiments [26], we found that a soft intensity thresholding on the images can improve the segmentation accuracy of U-NET segmentation. Next, we performed soft-intensity thresholding to remove pixels with image intensities less than 20 on a 255-intensity scale. Output segmentations may occasionally contain segmentation boundaries with blurred or fog-like regions that cannot be corrected by modifying the network. Since the majority of the segmented pixels fall in the 200–255 intensity range, and the blurred and fog-like regions have segmentation intensities in the range of 0–70, we applied a thresholding up to 70 to remove the blurred regions. Following thresholding, we binarized the segmentations to have all other bright pixels attain 255-pixel intensity. Once binarized, breaks in the inside morphology of each of these articulators may occur. To avoid such gaps, we have used dilation with a line structuring element of size 5 pixels to fill those gaps. Further, small independent regions on the segmentation map that are not relevant to the morphology of the vocal structures may occur (e.g., on the nasal airway, on the spine). Since segmentations of the individual vocal structures have a minimum number of pixels, we removed all independent segmentation maps with a total connected pixel count of less than 100 pixels. All pre-processing and post-processing functionalities were created using MATLAB R2021 (Mathworks, Natick, MA, USA).

Evaluation: In this study, we compared the performance of STL-UNET against the conventional U-NET model without transfer learning [26,28], and manual segmentation from the second human expert (vocologist). The segmentations from the first human expert (radiologist) were used as the ground truth. The following quantitative metrics were used in our comparisons:

1.DICE coefficient (D): this measures the spatial overlap between two segmentations according to the expression given below, where *A* and *B* are respectively the binary predicted segmentation and binary ground truth segmentation.
$$D=\frac{2|A\cap B|}{\left|A\right|+|B|}$$2.Hausdorff distance (HD): It is a distance-based metric widely used in evaluation of image segmentation as dissimilarity measures [33]. It is recommended when the overall accuracy (e.g., the boundary delineation/contour) of the segmentation is of importance. The HD distance between two finite point sets $A=\{a_{1},a_{2},\ldots ,a_{n}\}$ and $B=\{b_{1},b_{2},\ldots ,b_{n}\}$ is defined by.
$$HD=\max\;(h\left(A,B\right),h\left(B,A\right))$$
where *h*(*A*, *B*) is called the directed Hausdorff distance and given by:$$h\left(A,B\right)=\underset{a\in A}{\max}\underset{b\in B}{\min}\left|\left|a-b\right|\right|$$3.Segmentation count (Seg count): This is a metric used to count the number of segmentations generated by different methods. It is a useful tool for evaluating and comparing non-anatomical segmentations generated by the network. When segmenting the tongue or the velum, the ground truth value of Seg count should always be one. When segmenting the airway, Seg count can be >1, since the airway could be disjoint based on the vocal tract posture.

To establish a statistical measure while comparing pair-wise segmentations, the paired *t*-test was used for both Hausdorff distance (HD) and DICE metrics and the Wilcoxon paired test on the Segcount metric. We classified statistical significance at multiple *p*-values as: no significance—(*p* > 0.05), *—(*p* ≤ 0.05), **—(*p* ≤ 0.01), ***—(*p* ≤ 0.001), ****—(*p* ≤ 0.001).

## 3. Results

Figure 5 shows representative image frames from reference segmentation (Expert 1, radiologist), manual segmentation (Expert 2, vocologist), vanilla U-NET segmentation, and STL U-NET segmentation. For each protocol, two images from the test set were randomly chosen and shown. The segmentations in this figure are displayed with their corresponding mid-sagittal input image. DICE scores are provided for individual articulators when compared against the reference segmentation. We observe anatomically inaccurate segmentations largely in the Vanilla U-NET in comparison to the STL U-NET (see velum segmentations in Protocol 2, and airway segmentations in Protocol 1). This is reflected in the superior DICE measures of the STL U-NET in comparison to the vanilla U-NET. Segmentations between Expert 2 and Expert 1 showed similarities largely in images where boundaries were well defined (e.g., Protocol 1, 2; with DICE > 0.86). However, there was a lower DICE similarity (between 0.77 and 0.85) between the two experts in segmenting images from Protocol 3. This was largely due to increased air-tissue boundary blurring due to spirals at 3 T MRI. STL U-NET depicted similar trends as Expert 2, with similar DICE distributions. In contrast, vanilla U-NET showed overall lower DICE values and broken airway segmentations (e.g., see airway segmentations in Protocol 3).

Figure 6 shows the box-and-whisker plot representations of the DICE metric across the test images in each of the three protocols. In this figure, the columns represent the MRI protocols, and the rows represent the segmentation method. To help the reader evaluate the figure, a red dotted line is provided at a DICE score of 0.8, which typically indicates a high level of agreement. The graphs show that the vanilla U-NET had a wider distribution of values compared to the STL U-NET. In particular, in Protocols 2 and 3, the vanilla U-NET had DICE distribution of certain articulators less than 0.8. The tongue segmentations consistently provided good DICE scores for all three segmentation methods. The velum is the one articulator that showed the highest distribution of values among all methods. This may be due to two factors, (1) the difficulty of identifying the velar boundaries, and (2) the motion artifacts common in the velum during speech. The airway articulator had a wide distribution of DICE scores, but slightly less than the velum. The vanilla U-NET underperformed against both STL U-NET and expert segmentations by a great margin. Generally speaking, the distribution style of the STL U-NET graphs in Figure 6 is comparable to the expert segmentation.

Figure 7 is an aggregation of 27 graphs that compare three different metrics, three different articulators, and three independent MRI protocols. The performance of the network is evaluated on the grounds of (a) spatial performance using DICE score; (b) the ability to segment specific curvatures using a distance-based metric called Hausdorff distance; and (c) non-anatomical segmentation using methods that count the number of segmentations. Two types of statistical tests were performed between the three different segmentation methods. An asterisk “*” is used to indicate statistical significance when two individual bars are compared. Color coding indicates which method has superior performance. For example, in the case of HD distance, a lower value means better performance. Therefore, when comparing two methods, the asterisk will have the color of the method that has the lowest HD distance. In the case of the DICE score, however, the asterisk color indicates the method that had the highest DICE score. In the majority of HD distance cases, the vanilla U-NET segmentation values contained many outliers and higher average values, indicating poor performance. Out of nine comparisons between vanilla U-NET and STL U-NET, six had statistically significant differences. Generally, STL U-NET performed better than vanilla U-NET. In all other cases (with one exception), the STL U-NET and the inter-user variability (i.e., differences between Expert 1 and Expert 2) showed statistically better performance. The segmentation count metric allowed us to count the number of independent segmentations generated in each of the three methods. Segmentations generated by STL U-NET showed segmentation that counts are consistent with numbers from inter-user variability. This was not the case with the vanilla U-NET, which had segmentation counts inconsistent with expert segmentations. The DICE coefficient metric was used to compare the spatial overlap between the segments. In seven out of nine comparisons between methods, STL U-NET had a statistically significant DICE score compared with vanilla U-NET. When compared with inter-user variability, STL U-NET had similar or higher levels of performance, with the exception of a single case where expert segmentations performed slightly better against STL U-NET (DICE of velum in Protocol 2). In summary, when comparing these three independent imaging protocols, it can be concluded that STL U-NET provided statistically significant differences in most articulator segmentations against vanilla U-NET. When comparing the STL U-NET segmentation against the expert segmentation, in most of the cases, the STL U-NET had a similar performance. There were three cases where STL-UNET provided better segmentation than Expert 2 (DICE of the airway, tongue in Protocol 2, and DICE of the airway in Protocol 3). However, there were two cases where the expert segmentations outperformed the STL U-NET, and both were during velar segmentation in Protocol 2.

We conducted ablation studies to assess the effectiveness of the two pre-training datasets used in airway segmentation. Figure 8 presents a collection of bar graphs that compare the ablation cases and the STL U-NET. The results indicate that in all cases, the STL U-NET consistently performed in a manner similar to expert segmentations. When considering the HD distance and segmentation count metrics, the STL U-NET pre-trained with just Chest CT had the worst performance compared to the one pre-trained with the in-house airway MR dataset alone. Although the STL U-NET pre-trained with an in-house airway MR dataset showed very close results for Protocols 1 and 2, it diverged in Protocol 3, indicating inconsistencies across different protocols. On the other hand, the STL U-NET pre-trained with both Chest CT and in-house airway MR datasets showed consistent results in all three metrics and were comparable to expert segmentation. Figure 9 displays two samples from each protocol along with their ablation results. Visually, the segmentations from the STL U-NET pre-trained with either Chest CT or in-house airway MR datasets by themselves show significant non-anatomical segmentation. However, the STL U-NET pre-trained with both Chest CT and in-house airway MR datasets provided robust results across protocols, similar to the manual expert segmentations. 

## 4. Discussion

In this study, the authors developed a protocol adaptive stacked transfer learning U-NET model to automatically segment the vocal tract airspace, the tongue, and the velum in dynamic speech MRI. This approach leveraged low- and mid-level features from a large number of open-source annotated brain tumor MR and lung CT databases, and high-level features from a limited number of protocol-specific speech MR images. We showed utility in segmenting vocal structures of interest from three fast speech MRI protocols with different acquisition, reconstruction, and scanner field strength variations. When using a small training set of 20 images in each protocol, the STL U-NET model outperformed the vanilla U-NET model. There were no statistically significant differences in the majority of vocal structures in the segmentations from the STL U-NET model and human expert segmentation. The most significant differences were found in segmentations of the small structures surrounding the velum, where the transition between the soft tissue soft palate boundary and the bony hard palate boundary is fuzzy (e.g., see Protocol 2 HD and DICE metric comparison on the velum). We also observed lower metric scores in Protocol 3 in comparison to Protocols 1 and 2, largely because Protocol 3 employed a spiral trajectory at 3 T MRI and is more susceptible to off-resonance blurring at air-tissue boundaries. This was reflected in lower HD and DICE scores for all of the vanilla U-NET, STL U-NET, and Expert 2 segmentations during segmenting the airway and velum in Protocol 3. In this paper, segmentations from the proposed STL U-NET model were compared against segmentations from a vanilla U-NET model and segmentations from a human expert annotator. The vanilla U-NET model was chosen for comparison because it is the current state-of-the-art learning-based model in speech and upper airway MRI segmentation [28,29]. Comparisons against emerging networks, such as vision transformers and nn U-NET are beyond the scope of this initial feasibility paper and will be addressed in our future work. 

The introduction of large, annotated biomedical image datasets can greatly expedite the development of data-efficient neural networks customized for specific domains. Promising sources of such datasets include competitions, such as the Automated Cardiac Diagnosis Challenge [34] and the LAScarQS 2022: Left Atrial and Scar Quantification and Segmentation Challenge [35,36,37], which offer opportunities for pre-training neural networks in the field of biomedical imaging. Segmentation methods applied to the upper airway have the potential to aid in the investigation of carcinomas in the oropharyngeal, laryngeal, and tracheal regions, providing critical information on assessing speech dynamics and aiding the design of treatment plans. For example, in [38], the authors assessed speech in normal and post-glossectomy speakers using dynamic speech MRI. Articulation kinematics revealed by dynamic speech MRI could aid in the management and re-organization of speech production post-treatment of cancers in the upper airway regions (such as in post-glossectomy). There are a few noteworthy limitations to our study. First, we only segmented the tongue and velum as the major soft tissue articulators. In the future, we will extend this approach to segment other vocal organs, including the lips, epiglottis, pharynx, and glottis. Second, in this feasibility study, we tested this approach on ten randomly picked image frames from each of the three protocols, where these image frames were manually segmented by two expert annotators. We also randomly chose expert 1 (radiologist) as the reference segmentation. However, comprehensive testing warrants the manual annotations of multiple human experts. These human annotations are both time- and resource-intensive. We plan to extend our approach in future studies to (1) test a higher number of images, and (2) use methods, such as the STAPLE approach [39], to create reference segmentations from more than two human annotators. Third, the field of upper airway segmentation currently lacks large open-source datasets with annotated data, resulting in a lack of diversity in training data across sex, geography, and age. However, with advancements in dynamic time MRI technology, and access to dynamic speech MRI protocols on commercial scanners, and processing tools to assist humans in efficiently annotating the datasets, we anticipate the diversity of dynamic speech datasets will improve in the future. Another limitation of the STL U-NET model is that it uses three independent networks to segment the vocal airspace, tongue and velum, and does not leverage anatomical constraints. In the future, we will explore a unified network for joint segmentation of various vocal organs with implicit anatomical constraints (e.g., if the tongue moves backward, the oropharynx airway should be compressed). 

## 5. Conclusions

In this paper, we presented a stacked transfer learning U-NET model to automatically segment vocal structures in dynamic speech MRI. Our scheme leveraged low- and mid-level features from a large number of open-source annotated brain MR and lung CT datasets, and high-level features from a few protocol-specific speech MR images. In contrast to conventional vanilla U-NET, our approach is adaptable to multiple speech MRI protocols with different acquisition, reconstruction, and hardware settings. Using 10 randomly chosen test images from three different fast speech MRI protocols, our approach provided segmentations substantially better than the vanilla U-NET, and with similar accuracy as human experts, while needing very few datasets for re-training (of the order of 20 images). Future work will include comprehensive testing with multiple human annotators on a larger number of test images. Our method has the potential for significant savings in human time and resources for researchers and clinicians interested in the anatomy and physiology of the vocal tract. 

## Figures and Tables

**Figure 1 bioengineering-10-00623-f001:**
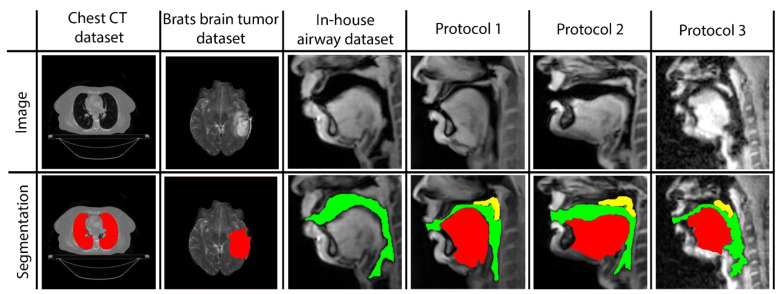
Samples of images and segmentations taken from all datasets used in STL U-NET training. MR images are provided in the first row, and their corresponding expert human segmentations are in the second row. Representative colors are used to identify the organ or articulator in the image. Tags representing each dataset are given as headings for each data instance.

**Figure 2 bioengineering-10-00623-f002:**
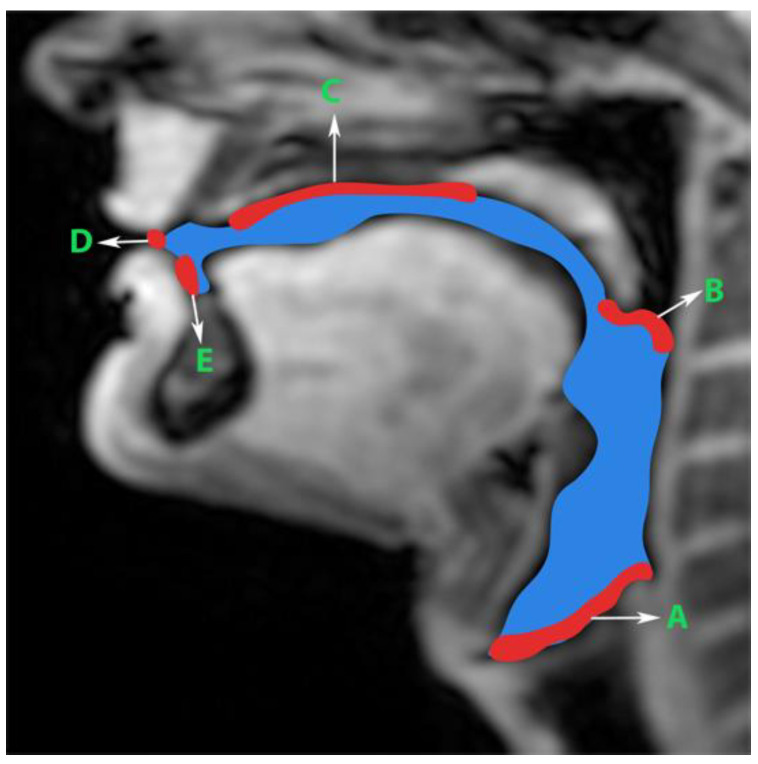
Prior to manual segmentation of the protocol speech MRI datasets, the anatomic boundaries of the vocal tract were established in consensus with all three human annotators: a vocologist, a radiologist, and a biomedical engineer. The blue colored region is the manually segmented upper airway. Red colored regions with notations show the boundaries for creating the manual segmentation. The boundaries are; A—lower boundary (vertebral column 6 and vocal fold), B—inferior boundary of velum, C—hard palate, D—a straight line connecting posterior edge of lower lip with upper lip, and E—region containing teeth up to the soft tissue connecting it.

**Figure 3 bioengineering-10-00623-f003:**
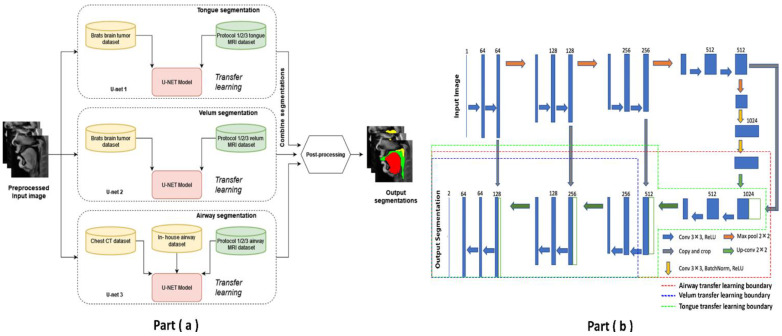
The stacked U-NET network architecture consists of three individual networks to respectively segment the tongue, velum, and vocal tract airspace. (**a**) provides high-level overview of network architecture with stacked individual U-NETs, describing the datasets used in training the respective networks. (**b**) shows the 39-layer U-NET architecture. Transfer learning enables sharing of weights from pre-training stage, and these weights are preserved as is in all the non-dotted layers. The dotted lines show the extent up to which re-training has been performed with protocol-specific speech MR data.

**Figure 4 bioengineering-10-00623-f004:**
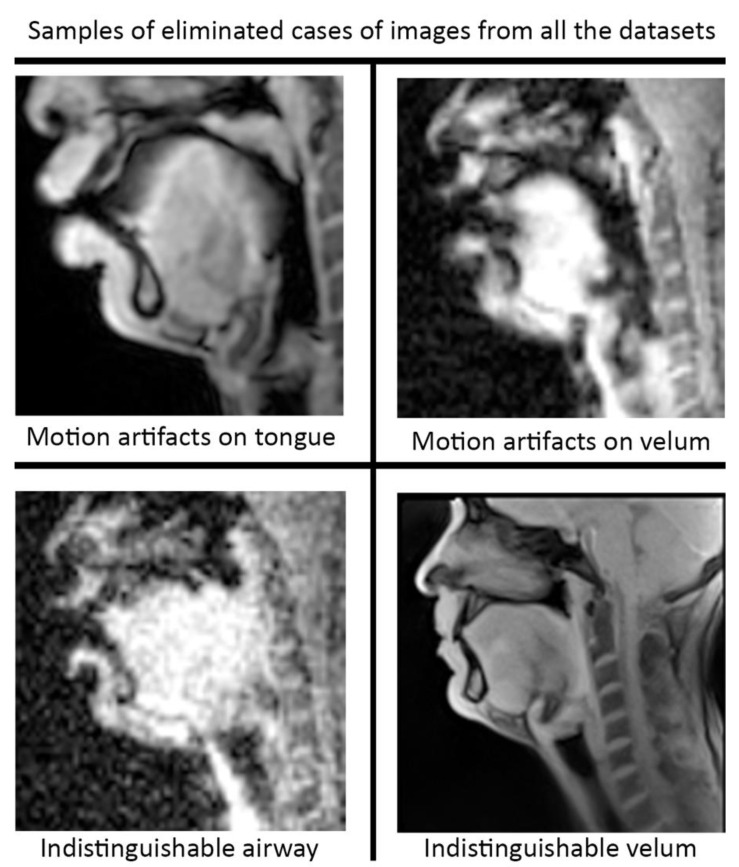
Data curation was applied prior to re-training the STL U-NET model. Image frames where the airway, tongue, or velum boundaries were blurred or indistinguishable. Shown here are examples of omitted frames from all the protocols.

**Figure 5 bioengineering-10-00623-f005:**
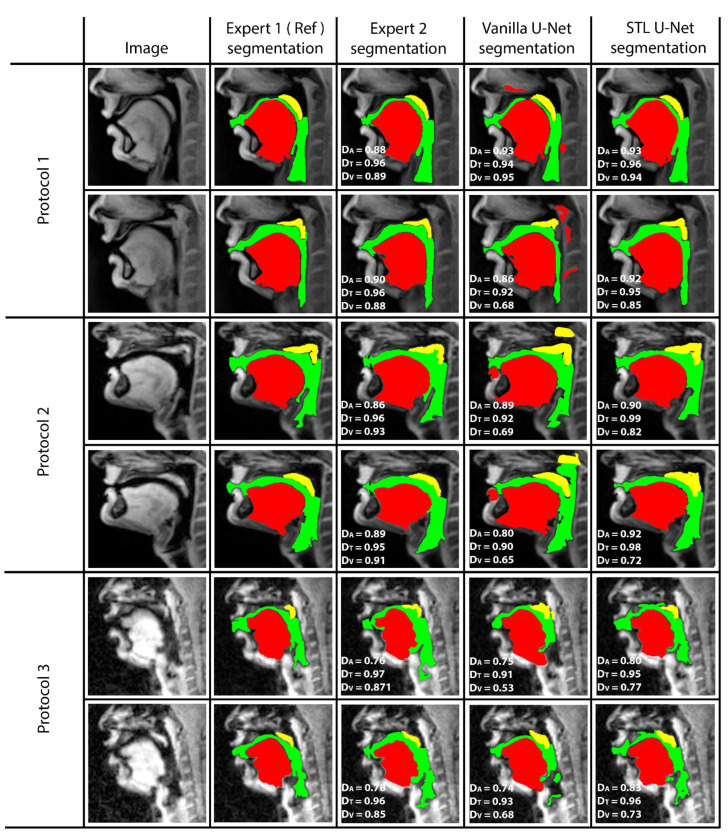
The figure shows two samples taken from each protocol and their DICE scores compared with reference. (1) column shows the sample image, (2) column shows the reference segmentation from Expert 1 (radiologist), (3) column shows the segmentation performed by the second expert annotator (vocologist), (4) column shows segmentation generated by vanilla U-NET architecture. (5) shows the segmentation generated by STL U-NET. DICE values are provided for all three articulators represented by different colors (Tongue: Red, Velum: Yellow, Airway: Green). Inter-observer variability can be observed by evaluating column (3), where two expert segmentations are compared.

**Figure 6 bioengineering-10-00623-f006:**
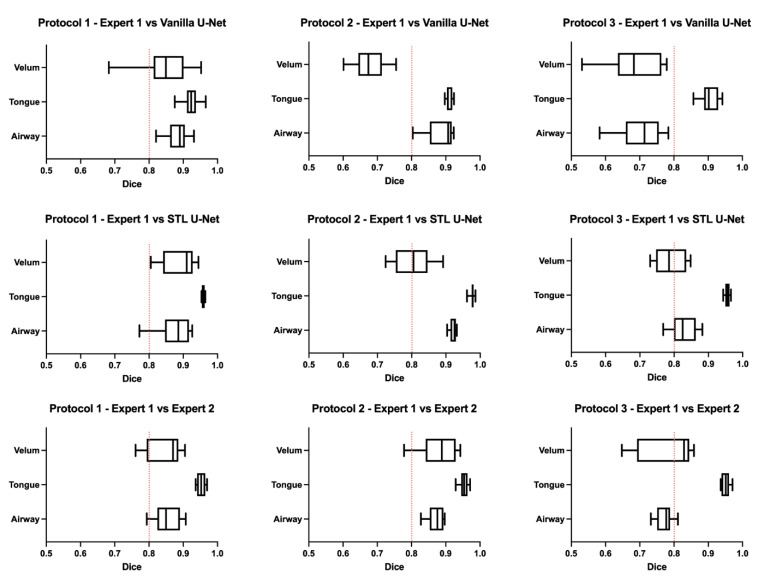
Box-and-whisker plots showing DICE value comparisons between (1) protocol-specific data from different MRI protocols used and (2) types of segmentation. The types of segmentations include (a) segmentation by Expert 2 (vocologist), (b) segmentations generated by vanilla U-NET, and (c) segmentations generated by STL U-NET. The plots are generated by using the entire test set for each of the protocols. A red dotted line is provided to show a 0.8 mark on the DICE coefficient for easily evaluating the performance of the segmentation modalities. The center line inside the boxes signifies the median of distribution; the width of the box corresponds to the interquartile range; and the left and right bars correspond to the minimum and maximum. One way to compare the segmentation performance of different protocols is by examining the individual columns that show each protocol’s various segmentation types. Upon observation, it becomes apparent that the distribution of STL U-NET is similar to that of Expert 2, whereas vanilla U-NET differs significantly from the expert segmentation.

**Figure 7 bioengineering-10-00623-f007:**
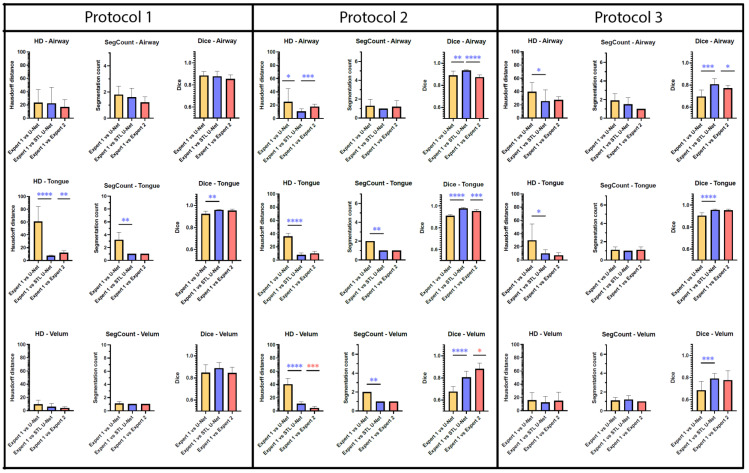
Bar plots are provided for all the three quantitative evaluation metrics (Hausdorff distance, segmentation count, and DICE coefficient) used for evaluating the segmentation, Whiskers show the outliers in data (as mean ± standard deviation). Paired *t*-test were performed for both Hausdorff distance (HD) and DICE; Wilcoxon paired tests were performed on segment count data. The significance of difference between each method is shown using “*“ marking which shows the *p*-value of the analysis. The correlation between the “*“ marking and *p* value is given as; no significance—(*p* > 0.05), *—(*p* ≤ 0.05), **—(*p* ≤ 0.01), ***—(*p* ≤ 0.001), ****—(*p* ≤ 0.001). Each primary column displays bar plots for individual MRI protocols, with sub-columns for different evaluation metrics. As we move down the rows, each row represents a specific articulator, such as an airway, tongue, or velum. Color coding has been given to identify best-performing segmentation method. Based on the observations, it is apparent that the STL U-NET model outperforms the vanilla U-NET model in most cases and is also comparable to expert segmentations. Interestingly, expert segmentations showed a better performance compared to both U-NET models in the case of Protocol 2 when evaluated using the HD distance and DICE score metric on the velum articulator. These findings suggest that the STL U-NET model is a promising approach for improving the segmentation accuracy.

**Figure 8 bioengineering-10-00623-f008:**
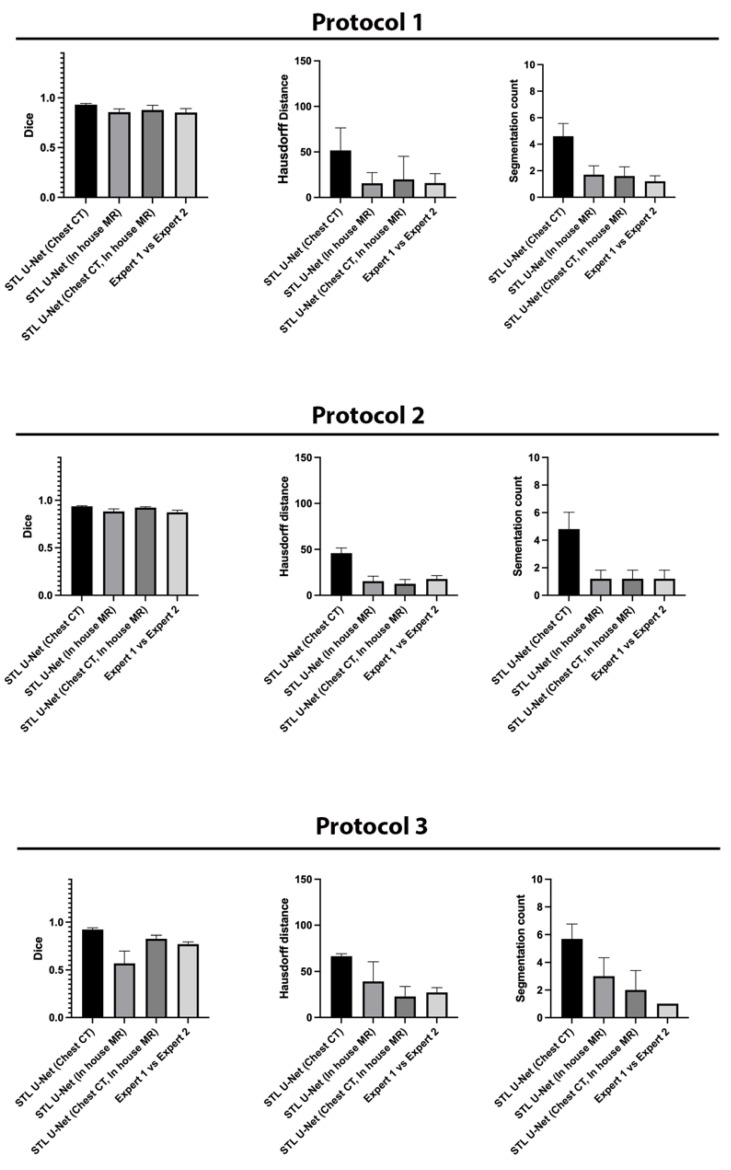
Ablation studies on the STL U-NET to evaluate its performance on airway segmentation using two different pre-training datasets: chest CT and in-house airway MR. Each pre-training dataset was removed during testing, and the results were compared against expert segmentation. The individual graphs display the ablation on the network along the x-axis; (a) STL U-NET(Chest CT): STL U-NET pre-trained with chest CT dataset and re-trained with protocol-specific data (Protocol 1, 2, or 3), (b) STL U-NET (in-house airway-MR): STL U-NET pre-trained with in-house airway MR dataset and re-trained with protocol-specific data, (c) STL U-NET(Chest CT, in-house airway MR): STL U-NET pre-trained with chest CT, in-house airway MR datasets and re-trained with protocol-specific data. The Y-axis represents the metric we used to assess the neural network's performance quantitatively. All three different protocols are tested, and results are provided under each sub heading and the inter-user variability is provided at the end for comparison on individual graphs.

**Figure 9 bioengineering-10-00623-f009:**
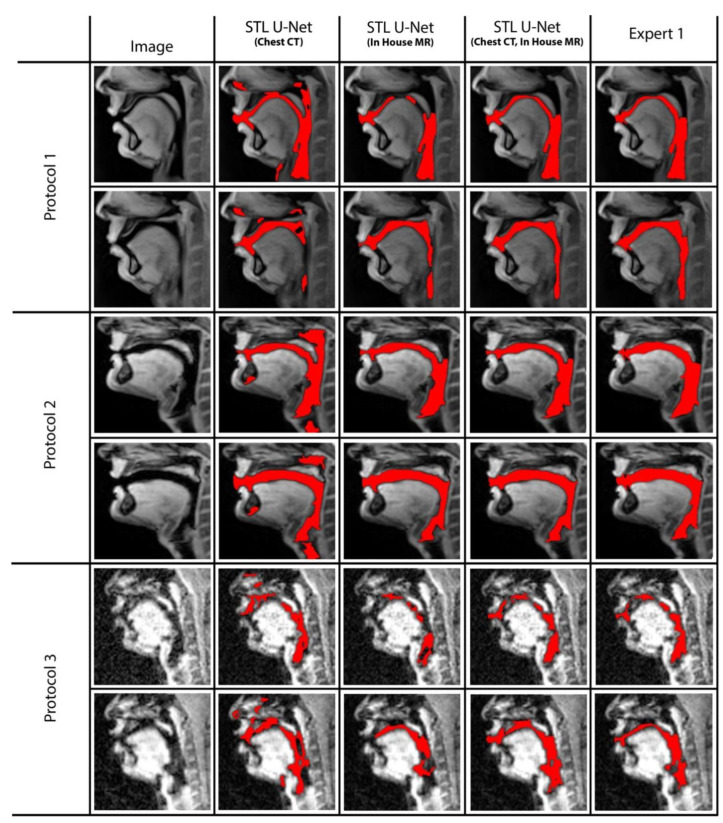
The figure displays two representative samples from each protocol, with the results of the ablation test overlaid on the image. Each row represents an individual protocol, and the columns show different ablations performed. Column (2) shows the STL U-NET (Chest CT), which was pre-trained with the chest CT dataset and re-trained with protocol-specific data (Protocol 1, 2, or 3). Column (3) shows the STL U-NET (in-house airway MR), which was pre-trained with the in-house MR dataset and re-trained with protocol-specific data. Column (4) shows the STL U-NET (Chest CT, in-house airway MR), which was pre-trained with both the chest CT and in-house MR datasets and re-trained with protocol-specific data. Finally, Column (5) shows the reference segmentation provided by Expert 1. Note that the performance of STL UNET improved by using both the in-house airway MR and chest CT datasets in pre-training as opposed to using only one of them. This is apparent in the reduced number of non-anatomical segmentations in the fourth column compared to the second and third columns. Particularly, note the closer representation of the STL UNET in the fourth column to the expert segmentation (fifth column).

**Table 1 bioengineering-10-00623-t001:** A list of hyperparameters used and the values considered when tuning the network. Grid-based search was implemented to try all the combinations from this list to be used for training the neural network. Fine-tuning is conducted after the best-performing combination is found after a grid search of this table.

Hyper Parameter	Values Considered
Number of epochs	60, 100, 200, 700, 1000, 1500
Steps per epoch	50, 100, 150, 200
Number of layers to re-train (training with protocol specific data)	3, 5, 10, 15, 20, 25, 30, 35
Learning rate	1 × 10^−4^, 3 × 10^−4^, 3 × 10^−5^, 1 × 10^−5^

## Data Availability

The source code and example data for the implementation and evaluation of our proposed STL U-NET is available here: https://github.com/eksubin/Protocol-adaptive-STL U-NET. Accessed on 1 December 2022.

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
