# Peer review of "Automatic Multiple Articulator Segmentation in Dynamic Speech MRI Using a Protocol Adaptive Stacked Transfer Learning U-NET Model"

_bioengineering, 2023, doi:10.3390/bioengineering10050623_

Round 1
Reviewer 1 Report
1. Did you do some hyperparameter optimization?
2. I recommend that authors should compare to the state-of-the-art methods, such as nnU-Net, Vision transformers, and so on.
3. You have introduced some previous work on 3D segmentation in the first part, but your approach seems to aim at 2D segmentation only. Is that make sense? In other words, if 3D segmentation has more practical significance?
4. Please discuss the results in a radiotherapy (application) context.
Reviewer 2 Report
This is an artical about clinical image segmentation. In the paper, it is presented a stacked transfer learning (STL) U-NET model to automatically segment vocal structures in dynamic speech MRI, inluding the vocal tract airspace, the tongue, and the velum in detail.
The condcuted strategy is to obtain low- and mid-level features from a large number of open-source annotated brain MR and lung CT datasets, and high-level features from few number of protocol specific speech MR images. The exerimental results show that the proposed approach providing segmentations substantially better than the vanilla U-NET, and with similar accuracy as human experts, while needing very few datasets for re-training. It could be improved in some of the aspects in the following.
1)In the experiments, the diversity and quantity of the verification data are not enough, inluding increasing more races, more different ages, and only in that way, it is convincing in wide applicatins and robust for the appoach.
2)The "transformer" is a well known model in deep learning and with good performance when applied in wide range modelling and one of mechanisms is the "attention", so is there any attention content in the STL U-NET? and why do you attempt applied transformer modelling in clinical image segmentation?
3)The comparison approach is only selected with the vanilla U-NET in the experiment, it is apparent that the proposed method is better in performce since no adaptive training is on vanilla U-NET, so why didn't you compare the experimental performance with the other related algorithm, say reference[33]?
Reviewer 3 Report
This paper titled “Automatic multiple articulator segmentation in dynamic speech MRI using a protocol adaptive stacked transfer learning 2 U-NET model” presents an interesting work on speech MRI segmentation, However, I’d like to raise some issues,
1. First of all, it is just a direct application of the U-Net model on the speech MRI segmentation, and the theoretical novelty is very limited.
2. The authors adopt the strategy of pretraining the U-Net model on the brain tumor MR and lung CT datasets, and an in-house airway labelled dataset, why just three datasets are employed for pretraining? How about the results if less datasets or more datasets are employed for pretraining? Suggest the authors take the BiVentricle dataset and the LA dataset into account, in the following links and works,
(1) https://acdc.creatis.insa-lyon.fr/description/results.html Biventricle dataset
(2) https://zmiclab.github.io/projects/lascarqs22/ LA dataset and those in the following work,
(3) Context-aware Network Fusing Transformer and V-Net for Semi-supervised Segmentation of 3D Left Atrium, https://doi.org/ 10.1016/j.eswa.2022.119105
3. Usually, there should be some ablation studies to show the performance of each module in the proposed model, however, it is just a direct application of the U-Net, so, more experiments are required to verify the effectiveness of each dataset for pretraining. If more pretraining datasets lead to better results, how to determine the amount of pretraining datasets?
4. The experimental results are not compared with other SOTA methods, such as transformer based methods, such as the one in the above ref. , or others, such as TransUNet, U-Net Transformer, Swin-Unet, TransAttUnet, Mask2Former… since these methods outperforms the U-Net, it is expected that the results will be better if the U-Net in the proposed method is replaced by these SOTA methods.
5. It is also necessary to verify the proposed strategy by comparing with other methods specific to speech MRI segmentation.
Reviewer 4 Report
The authors proposed a stacked transfer learning U-NET model to segment the deforming vocal tract, the tongue, and the velum in 2D 15 mid-sagittal slices of dynamic speech MRI. The proposed approach used mid-level and high-level features. The proposed approach is demonstrated using three protocols such as Protocol 1 : 3 T, Protocol 2: 1.5 T and Protocol 3 : 3 T 22.
Comments to authors:
i. Include the contribution and novelty at the end of the Introduction.
ii. Add the supporting related work to understand state of the art better.
iii. Include the working procedure or algorithms for better understanding at the end of section 2.
iv. Give the summarized implementation datasets in Section 2 or Section 3.
v. Include more information about Figures 6 and 7 for better understanding.
Round 2
Reviewer 1 Report
The authors have addressed some of the existing issues. However, even if the most advanced networks have not yet been applied in this field, it is more meaningful to conduct comparative experiments using the most advanced methods.
Author Response
Thanks for the comment. We have modified our statement in the discussion section to reflect such comparisons are needed in future work.
"Comparisons against emerging networks such as vision transformers and nn U-NET are beyond the scope of this initial feasibility paper and will be addressed in our future work."
Reviewer 4 Report
The authors are addressed my all comments.
Author Response
Thank you for the response, we appreciate it.